# Monitoring Particulate Matter with Wearable Sensors and the Influence on Student Environmental Attitudes

**DOI:** 10.3390/s22031295

**Published:** 2022-02-08

**Authors:** Frances Kane, Joseph Abbate, Eric C. Landahl, Mark J. Potosnak

**Affiliations:** 1Department of Environmental Science and Studies, DePaul University, Chicago, IL 60614, USA; franceskane96@gmail.com (F.K.); jmabbatejr@gmail.com (J.A.); 2Department of Physics, DePaul University, Chicago, IL 60614, USA; ELANDAHL@depaul.edu

**Keywords:** air quality, wearable sensors, particulate matter, environmental attitudes, community science

## Abstract

The mobile monitoring of air pollution is a growing field, prospectively filling in spatial gaps while personalizing air-quality-based risk assessment. We developed wearable sensors to record particulate matter (PM), and through a community science approach, students of partnering Chicago high schools monitored PM concentrations during their commutes over a five- and thirteen-day period. Our main objective was to investigate how mobile monitoring influenced students’ environmental attitudes and we did this by having the students explore the relationship between PM concentrations and urban vegetation. Urban vegetation was approximated with a normalized difference vegetation index (NDVI) using Landsat 8 satellite imagery. While the linear regression for one partner school indicated a negative correlation between PM and vegetation, the other indicated a positive correlation, contrary to our expectations. Survey responses were scored on the basis of their environmental affinity and knowledge. There were no significant differences between cumulative pre- and post-experiment survey responses at Josephinum Academy, and only one weakly significant difference in survey results at DePaul Prep in the Knowledge category. However, changes within certain attitudinal subscales may possibly suggest that students were inclined to practice more sustainable behaviors, but perhaps lacked the resources to do so.

## 1. Introduction

Air quality is a global concern that imposes a direct threat to the health of those within the urban environment and does so in a way that disproportionately impacts the marginalized communities of the industrialized world, as they are most often in closer proximity to threatening emission sources [1]. Coming from a range of stationary and mobile sources, such as fuel-combusting power plants and on- or off-road vehicles, respectively, air particulate matter (PM) is a very common concern within cities. PM refers to a mixture of organic and inorganic particles or liquid aerosols that, upon inhalation, have been known to cause serious obstruction to heart and lung function for those with long-term exposure or preexisting conditions such as asthma [2]. Studies have shown that even low levels of exposure to air pollutants such as PM can have adverse effects on human health [3]. In order to mitigate the hazards of PM and other urban air pollutants, it is important to have an understanding of when and where there is a high risk of exposure [4].

Today, air quality data, including those of PM, are predominately collected through large and expensive stationary monitors [5]. The use of remote sensing instrumentation is another growing approach. In the past several years, remote sensing technology—particularly through Internet of Things (IoT) technology—has become increasingly accessible [6,7,8]. While it is highly effective at observing the spatial distribution of air pollutants on global and national scales, these instruments are currently limited in their capacity to obtain more localized, near-surface readings. [9]. Furthermore, through these current approaches, there is not a direct connection between archived observations and the general public. While the US EPA has AirNow, the information provided is relatively coarse [X]. As a result, most individuals do not understand the health concerns specific to their own surroundings and how their exposures may differ from the exposures experienced by others. Wearable air quality sensors can fill this gap [10].

Moreover, there is interest in expanding on the modes through which atmospheric data are collected. One such method may be the extensive mobile monitoring of air through community science: a model of scientific investigation that incorporates the public in the processes of posing questions, collecting data, and interpreting and communicating results. A project conducted by Kansas State University examined this specifically in the context of environmental justice (EJ) communities in Chicago, Illinois, in which residents were given low-cost air pollution sensors to collect PM data in their neighborhoods [11]. With air quality monitoring tools directly in the hands of the general public, the implications of air quality data should resultantly become a more personal matter. By bringing these observations to a more localized scale, concerned individuals or communities are better equipped to bring matters of environmental health into an actionable and political context [7,12,13,14]. The Luftdaten project found that IoT/PM monitoring has increased visibility and utility of PM data amongst more media coverage on local air quality issues in Stuttgart, Germany [14]. On the other end of this collaborative process, scientists could more thoroughly study the social and geophysical dynamics of air pollutants in areas where stationary monitors may not currently exist.

Successful “crowdsourcing” projects, such as the Environmental Monitoring Assessment Network (EMAN)’s NatureWatch, and the Cornell Lab of Ornithology and National Audubon Society’s eBird, have demonstrated how the scientific community can promote education and awareness on ecological issues while concurrently expediting the process of widespread data collection [15]. Research from Bouvier-Brown [16] furthers this notion, implying that individuals develop a much stronger connection to air pollution and other issues of environmental justice when they engage in hands-on learning with community members [17,18]. Bouvier-Brown also addresses the lack of affordable mobile monitoring devices, and the promising future they may have in the field of community-science-based research. Since 2014, accessible mobile monitoring devices have become more widespread. The Environmental Protection Agency (EPA) is rolling out the Los Angeles Library Loan project, in which residents can loan out portable air sensors from public libraries [19]. In addition, the EPA released the AirMapper, a portable air sensor that includes a GPS and can collect data on various pollutants such as particulate matter and carbon dioxide [20]. There are also emerging data-science toolkits that can assist with the visualization and analysis of wearable air quality data [21]. New-generation sensor designs exist that can integrate geospatial referencing and wireless data communication [18], but they do raise privacy and data-ownership issues.

In this study, we predicted that engagement with wearable sensors would cause a heightened environmental affinity and sense of responsibility for issues surrounding anthropogenic pollutants; and additionally, that data acquired on these pollutants can be integrated with other forms of data to further elucidate matters of public and ecological health. Some research has been established in forecasting the potential of these intersections—those of which may include air-quality-selective rerouting and other real-time mobile alert services for map applications [12,22]. One such application, PulsAir, has already shown how a questionnaire-driven risk assessment (re cardiovascular disease, type 2 diabetes, asthma) can be further contextualized for the user through its integration with localized environmental risk factors and certain wearable technologies (e.g., FitBit) [12]. With this in mind, our study examined the social and environmental implications of mobile monitoring.

Students of partnering Chicago high schools were able to take the mobile monitoring of PM into their own hands, posing questions, collecting data, and interpreting their findings. They were thoroughly instructed on the process of collecting data through the wearable sensors and logging spatiotemporal observations. Collectively, we investigated the relationship between Chicago’s concentrations of PM and urban vegetation. Pre- and post-experiment surveys were conducted to assess the influence of mobile monitoring on participating students’ environmental attitudes and knowledge.

Ultimately, two hypotheses were tested: (1) Students’ scores on the post-experiment survey instruments will indicate a heightened environmental affinity and knowledge, and (2) PM concentrations will be lower in more vegetated study areas. In an effort to limit any respondent bias in survey performance, the details of the affinity and knowledge hypothesis were temporarily withheld from students. Upon completion of the post-experiment surveys, this role of deception was disclosed to all participants.

Addressing the first hypothesis that considers the environmental attitudes of students requires the tools of social science, while the second hypothesis on air quality and vegetation is based in the natural sciences. Testing the first social science hypothesis places a number of limitations on the methodology used for the second hypothesis.

Working with these partner high schools and the students, some of whom were minors, imposed significant logistical constraints on the experiment. Since our goal was to test the efficacy of a participatory experiment on student environmental attitudes, the students were officially considered human subjects and approval was necessary from DePaul University’s Institutional Review Board. In addition, we needed consent and approval from the following groups at two different high schools: administrators, teachers, parents, and of course the students themselves. This consent was particularly difficult to secure in the city of Chicago, because of its history of environmental racism. Many community members, and particularly community members of color, are suspicious of data collection efforts.

We made several design choices about the sensors to gain the trust and approval of these groups. First, all data were collected via a storage card and had to be manually downloaded. To preserve data integrity, no wireless interaction with the devices was possible. Second, no locational data were collected and no GPS units were used. All location information was extracted from hand-drawn maps that the students themselves recorded. This hard-copy process served as an additional layer of informed consent. Moreover, the process of building trust and gaining consent was time consuming, so only two classrooms were studied over several years. Each class was relatively small and so the overall sample size is small.

## 2. Materials and Methods

The experiment was completed twice at two different high schools in the city of Chicago and a similar protocol was conducted each time. First, students took a pre-experiment survey. Next, the students learned about air quality and then used the wearable sensors to measure air quality near their home. Students picked up the sensors each day before leaving school and then returned the sensors the next day. The sensors were charged and the students recorded their movements on the map each day. After the experiment ended, students took the post-experiment survey.

### 2.1. Surveying Procedures

The two study schools were Josephinum Academy of the Sacred Heart and DePaul Prep High School. Josephinum Academy is an all-girls high school in Chicago, Illinois’ Wicker Park neighborhood, and DePaul Prep High School is a Catholic high school located in Chicago, Illinois’ Irving Park East neighborhood. At Josephinum, nine students of a predominately senior-level environmental science course agreed to participate. At DePaul Prep, 22 students from a chemistry class agreed to participate. For both schools, the first step was to complete a pre-experiment survey. For the purpose of confidentiality, each student was administered a sequential ID code from their teachers to eventually link the results of their initial survey with that of their PM sensors and final survey.

The survey instrument was an adaptation of the Children’s Environmental Attitude and Knowledge Scale (CHEAKS) [23]. The original 66-question instrument was selectively narrowed down to 20 questions, which focused the survey primarily on issues surrounding anthropogenic pollution. Both the adapted and the original CHEAKS surveys (see Appendix A) split evenly into four subscales, assessing verbal commitments, actual commitments, affect, and knowledge. The former three criteria were measured on a 5-point Likert-type scale, ranging from “(1) very true,” denoting firm agreement with a statement, to “(5) very false,” denoting firm disagreement. Knowledge-based questions, however, were strictly objective (e.g., “Most of the lead in our air is caused by:”).

Using a protocol and instrument approved by DePaul University’s Institutional Review Board, all pre-experiment surveys were completed and collected at the beginning of the initial classroom intervention. Following the period of PM data collection, the same CHEAKS instrument was distributed as a post-experiment survey. Upon completion of both instruments, all students were debriefed on the role of the surveys, and how their responses would be processed to study the changes on environmental attitudes through wearable sensing.

For the 15 questions of the Likert-type scale related to attitude, students’ responses were scored 1–5, with scores representing the least and greatest environmental affinities coded as “1” and “5”, respectively. A one-tailed paired *t*-test was run for the survey sets of both schools, pairing each student’s cumulative pre-experiment survey score with their cumulative post-experiment survey score (n = 9 for Josephinum, and n = 19 pairs for DePaul Prep since two surveys were incomplete and one was missing for the 22 students). For the five knowledge-based questions, responses were scored with a binary “0” or “1” code representing a correct or incorrect answer. The same one-tailed *t*-test procedure was run for these responses, pairing each student’s total of correct responses from pre-experiment survey to post-experiment survey. A one-tailed *t*-test was run because our a priori hypothesis anticipated that students would have a heightened environmental affinity in the post-survey.

### 2.2. Mobile Monitoring, GIS, and Remote Sensing Procedures

While there are state-of-the-art techniques to record position and wireless transmit data [24], gaining permission to perform the study with high school students created challenges. Many of the participating students were minors (under age 18) and the permission of their parents was necessary. To alleviate privacy concerns, the sensors were designed with no wireless interface and no GPS or other location information. The sensors stored data on a microSD card that needed to be downloaded manually. Students recorded their movements with a low-tech paper map and pencil; a real-time clock on the device was used to synchronize concentration data with the student’s location recorded on the map with timestamps. Chicago has a long history of environmental racism [25,26], and all experiments involving community partners and particularly children need to be designed very carefully to preserve the rights of participants. By involving community partners in the collection and analysis of data, and by ensuring anonymity by using sensors that do not collect GPS information, we were able to create a foundation of trust that may not always be present in scientific studies [27,28]. Future efforts could consider blockchain technologies being pioneered in healthcare that could preserve patient privacy and control and verify data access [29].

Wearable PM sensors were designed using existing open-source hardware designs and software programs. The sensors (Figure 1) detect the presence of particulates through a dynamic light scattering technique. When turned on, the optical devices pulse a beam of infrared light once per minute and record a time-stamped PM concentration based on the total amount of reflected IR light. These measurements are recorded in volt units that are not standardized in common PM units. Though the sensors do not distinguish between PM classes (e.g., PM_2.5_, PM_10_), they are designed to monitor cumulative concentrations linearly across all particulate size ranges. While these PM sensors have limitations related to cross sensitivities with humidity and temperature and also cannot detect fine-mode aerosols, their low cost and portability are ideal for the present experiment [30]. Moreover, their principle of detection is simple enough to be understood with a high school science background.

Though marketed for detecting dust (Sharp GP2Y1010AU0F), these sensors have been used in a number of other air quality projects. The sensor data were logged through a microcontroller with a microSD card interface from Adafruit (New York City, NY, USA), containing a real-time clock on a daughter board (Adafruit Feather 32u4 Adalogger, Product ID: 2795; DS3231 Precision RTC FeatherWing—RTC Add-on for Feather Boards, Product ID: 3028). A simple resister (150 Ω) and capacitor (220 µF) circuit buffered the increased power requirement when the infrared LED was fired briefly (~350 ms). To reduce noise, the LED was lit, measured, and turned off 10 times, approximately one cycle per second, every minute. The 10 sequential readings were averaged to produce one measurement per minute. A small (3.7 V, 350 mAh, Product ID: 2750) LiPo battery provided power. There was no location information (GPS) stored and all data needed to be downloaded manually via the microSD card. This was necessary for privacy concerns around gaining permission for using high school students as human test subjects. The sensors were housed in plastic project boxes. While not used for the present experiment, a subsequent project with DePaul Prep used the same sensors with a 3D-printed case designed by the students.

Students were introduced to our hypothesis addressing PM concentrations and urban vegetation, and were encouraged to consider hypotheses of their own over the several following weeks. A standardized method of mobile data acquisition was covered. This involved each student attaching their sensor to a strap of their backpack with a carabiner clip at a height above ground of about 1 m. Establishing this standard design would increase airflow through the device’s aperture while walking, and minimize the likelihood of some students collecting more airflow than others. The device had no active airflow. The measured concentration field of urban sensors is very complicated and complex, but a length-scale of 500 m provides satisfactory results [31].

The PM data collection took place over a five-day period for Josephinum and a 13-day period for DePaul Prep. For both schools, students collected PM data for their commutes between school and their homes, recording the times in which they had exited and entered, respectively. Each student was given a physical map of the study area, in which they provided their routes for each day by hand drawing them on the map. On the final day, sensors were collected upon entering the school. With a 24 h battery life, the sensors continuously collected PM concentrations once every minute with the exception of school hours, in which they were each connected to a charging port. Students were instructed to drop off and pick up their devices for this daily charging.

Because modes of commute varied and again being sensitive to privacy issues, sensor PM data were only considered entering and exiting student homes and the school. Based on the students’ spatiotemporal data logging on the paper maps, a total of 93 time and location pairs were determined for eight different geographical locations for Josephinum: Josephinum Academy itself and the homes of students wearing sensors 1 and 3 through 8. PM readings for sensor 2 and sensor 9 were discarded due to insufficient data logging. For each of the 93 pairs, a mean average of PM concentration was recorded over an 11 min window (that is, 11 readings). This window minimized the chances of recording sampling error while also focusing our analysis on the periods when students entered or exited their home or the school. Thus, the PM values used in our analysis for both the “home” and “school” sites were the mean average PM values recorded in this 11 min window. Our approach excluded indoor PM concentrations (i.e., at home, at school, in cars, buses, or trains). While indoor PM is an important contributor to an individual’s exposure to air pollution, accounting for it would introduce many unique challenges to the spatial component of this analysis. Moreover, extremely high values, for example, due to cooking or cigarette smoke, would highly skew the data. In summary, the wearable sensor time series data were reduced to point measurements with the following steps:Students wore sensors during their commute to and from school each day.Students recorded their commute to and from school on a paper map.Students recorded the start and end times of their commute.Data were downloaded from the wearable sensor storage cards.Student routes were matched to sensor IDs.The first 11 min and the last 11 min of the commute were collected and averaged using the sensor timestamps from the real-time clocks and the start and end times recorded by students.Each of these averaged values was considered a time and location pair.These time and location pairs were subsequently averaged to derive a concentration value for the student’s home location.Using a much larger set of pairs, concentration was determined for the high school from multiple students’ sensors.

As for DePaul Prep High School, a total of 69 pairs of time and location were determined for eight different geographic locations: in this case, DePaul Prep High School as well as the homes associated with sensors 1–3, 5–6, 8, and 10. Similar to Josephinum, PM readings for sensors 4, 7, and 9 had to be discarded due to insufficient data logging. For the DePaul Prep data, the mean average of PM concentration was again recorded over an 11 min window. For both data sets, we calculated an average PM concentration for each site (student homes + school). Note that these analysis steps were kept relatively simple so that the process could be explained to the student participants. This was necessary to adequately test the environmental attitudes hypothesis. Students needed to understand the process as full participants.

For the geospatial analysis of the route data collected from the students, a set of coordinates was approximated for each monitoring site and plotted as point features in ArcMap. For each site, a zone of a half-kilometer radius (500 m) was created for determining vegetation cover. We used Landsat 8 combined Operational Land Imager and Thermal Infrared Sensor (OLI/TIRS) imagery. The imagery was selected based on minimal cloud cover and its proximity to the approximate time frames of the wearable sensor data acquisitions (September–October, 2016 for Josephinum and February–March, 2017 for DePaul Prep). In order to determine the relative amount of urban vegetation, we used a true normalized difference vegetation index (NDVI) in which 0.5 < NDVI < 1 indicates a high level of vegetation, and NDVI < 0.140 indicates little to no vegetation [32]. Accounting for the intensities of visible red and near-infrared light (NIR) reflected by vegetation, this index calculates the density of a landscape’s foliage through the spectral differencing formula:(1)$$\mathrm{NDVI}=\frac{\left(\mathrm{NIR}-\mathrm{RED}\right)}{\left(\mathrm{NIR}+\mathrm{RED}\right)}$$

Spaces of healthy vegetation in our study area should have absorbed a greater amount of visible red light (around 650 nm) and reflected a greater amount of near-infrared light (700–1100 nm). By overlaying the NDVI and zoned PM point features, we used the NDVI value to define the amount of vegetation for each of our 14 study sites across both schools. We ran a linear regression on these data, with the sample of our viable monitoring sites (n = 7 for each), modeling the changes in PM concentrations in response to our tabulated areas of vegetation.

## 3. Results

### 3.1. Survey Data Analysis

Based on the standard alpha level, α = 0.05, we found no significant differences between students’ responses to the pre- and post-experiment survey instruments for Josephinum Academy (Figure 2 and Table 1). This was the case for the assessment of overall environmental attitudes at Josephinum (*p* = 0.65), as well as that of environmental knowledge (*p* = 0.50). For the most part, this was the same for DePaul Prep (Figure 2 and Table 2), with the only significant difference in survey results in the Knowledge category, where students’ knowledge of environmental issues slightly increased (*t* = 0.04). Although the pre- and post-experiment survey responses did not yield any significant differences for Josephinum, the consistency between the mean scores of these surveys does seem to indicate that the CHEAKS instrument serves as a robust assessment tool for studying the variables of our qualitative hypothesis. Had we run a two-tailed *t*-test, the survey results at DePaul Prep would have revealed a significant but negative difference in the Verbal category within attitudes. The reason the Verbal category is not significant is because the nature of the one-tailed *t*-test can only show an increase or heightened affinity for environmental issues, in accordance with our initial hypothesis.

A potential source of error may have been the sequencing of our methodology. The interpretation of data and the process of determining results is an important aspect of the community science process that was not completed before the post-experiment survey. While having students be both experimental subjects and also researchers can raise complex ethical questions [33], true community science involves participants in all phases of research. Although this inclusion may create issues around data integrity and reporting bias, revealing the results of the PM analysis prior to the post-experiment survey could significantly influence students’ responses. In addition, the distribution of the survey toward the end of the school year at DePaul Prep could have had an influence on student responses on environmental attitudes. This might explain the tendency at DePaul Prep in the Verbal category where students expressed more negative and less “enthusiastic” attitudes toward the environment. There is also the possibility that this was a type II (false positive) error. Expanding considerably on the sample size of n = 9 and n = 19 could help minimize the likelihood of type II error, and overall reveal more insightful evidence of the relationship between wearable sensors, environmental affinity, and the knowledge-based response to anthropogenic pollution.

In building on this study, there may also be implications of the changes in response (or lack thereof) to particular types of survey questions. After completing the five days of air quality monitoring, 5 of the 9 respondents at Josephinum Academy indicated a heightened affirmation that, “[they are] frightened to think people don’t care about the environment.” Concurrently, only 2 of the 9 respondents indicated a heightened affirmation that they “would be willing to ride the bus to more places in order to reduce air pollution”. Such a trend may suggest that these students are well inclined to practice more environmentally conscious behaviors, but perhaps, are lacking the resources to do so. This pattern was similar at DePaul Prep, as over half of the students expressed high concern over the effects of pollution on their families. Between the pre- and post-experiment survey, the number of students expressing this concern went from 11 to 13. A deeper analysis with a greater sample size will be required to adequately investigate the relationship between a student’s behavioral responses to changes in environmental affect.

In addition to gathering a greater sample size, a more representative sample group may yield more replicable results. Since the sample group at Josephinum Academy all belonged to an environmental science course, these subjects may have had broader knowledge of environmental issues than their peers, and perhaps had stronger opinions about them. While still focusing on youth, this study could be supplemented by expanding to classrooms and after-school programs of different subjects. To thoroughly understand how the general public responds to the mobile monitoring of air quality, continued research should also incorporate intergenerational sample groups.

In further studies, it may also be of interest to analyze whether or not these heightened affinities of environmental affect withstand or regress over time; and furthermore, if actual commitments have a tendency to build over time, acknowledging that several students expressed an increased verbal commitment without any evidence of increased actual commitment. Similarly, there could be significant changes across all attitudinal subscales when extending the periods in which participants collect air quality data. With these concepts in mind, future approaches to this project may involve increasing the number of days that students collect PM data, the chronology of survey data collection and presentation of results, and perhaps, the addition of a third survey phase after a greater period of reflection and habit formation.

### 3.2. PM Data Analysis

As we had hypothesized, our correlational analysis of PM concentrations and NDVI values for Josephinum suggested an inversely proportional relationship, in which PM concentrations were lower in more vegetated study areas (Figure 3). However, the data from DePaul Prep show the opposite relationship. According to our linear regression with an r^2^ value of 0.43 for Josephinum and 0.46 for DePaul Prep, our linear model indicates vegetation presence can explain nearly half of the variation in PM concentration.

Through a more generalized observation of our broader study area, the inverse proportionality of PM and NDVI appears to hold true, in part. As mentioned, in Figure 3 we notice that the monitoring sites with the highest NDVI values, in the case of Josephinum, have lower PM concentrations. Considering Figure 4, many of the Josephinum monitoring sites are either located in suburban areas, or areas in the city with more vegetation. On the other hand, many of DePaul Prep’s monitoring sites are located in more urban areas, where there tends to be less vegetation.

As shown in Figure 3, DePaul Prep’s data demonstrate a positive correlation, which is in opposition to our hypothesis. Perhaps these results tell us that it is more difficult to quantify vegetation with such a small radius of 0.5 km. Looking at Figure 4, the urban landscape seems to be rather varied, and the fragmentation and low-density vegetation may make it difficult for a small radius to pick up on these greater patterns. Moreover, many of the DePaul Prep locations were near transportation corridors, which we speculate could disrupt the expected pattern between PM and vegetation.

## 4. Discussion

These observations also provoke a number of questions exterior to NDVI—one of which may pertain to the transportive nature of PM. Regardless of an emission’s origin, it is important to note that, within the urban landscape especially, previously settled PM is susceptible to resuspension through vehicle-induced turbulence [34]. Although the residence time of tropospheric PM does not tend to exceed a few days to a few weeks, a more temporally extensive approach may be necessary to standardize the interference of the atmospheric advection and even trans-boundary potential of particulates [35,36]. For similar reasons, it would be highly beneficial to expand PM observations to a broader range of vegetation types and densities. Perhaps a classified analysis may reveal that certain types of vegetation suppress the suspension of PM more effectively than others [37].

Finally, it is also important to consider that not all outdoor air pollutants—PM included—should be attributed to clusters of population and industry. We should also question how and why air pollutants independently persist in less urbanized spaces. As the accessibility and compactness of mobile monitoring devices continue to advance through research and technology, answers to such questions, as well as potential modes of civil action, should grow increasingly transparent.

While our study did not find that participation in the community science project increased environmental attitudes, wearable sensors are still a valuable tool for addressing environmental justice issues. The city of Chicago continues to see air pollution as an environmental justice issue. While the closing of a coal-fired power plant within the city limits was a success, the implosion of the smokestack during the decommissioning process spread particulate matter over a community of color. This lead to accusations of environmental racism and also highlighted the lack of fine-scale measurements [38]. Moreover, factory locations within the city of Chicago highlight further tensions over environmental justice. General Iron is a metal recycling company that operated for many years in Lincoln Park, a majority-white neighborhood. However, the riverfront real estate was being developed for residential and commercial use, so General Iron sold the property and moved to the city’s southwest side, a neighborhood of color [39,40]. The air pollution is the primary concern, from both operations and from increased heavy-vehicle traffic. Hazardous air pollutants are also a concern. The Sterigenics facility in Willowbrook, a suburb of Chicago, sterilizes medical equipment but releases ethylene oxide. This has spurred a series of lawsuits [41]. These recent cases illustrate the concern focused on air quality and how emission sources are often located in neighborhoods of color. Wearable air quality sensors have an important role in providing community members with actionable steps to identify sources.

An area for future research is using blockchain technology to secure and authenticate data gathered by community partners with wearable sensors, as has been proposed in the wearable medical device field [29]. Communities of color are often distrustful of outsiders due to a history of racism and economic underinvestment. Our present study was limited in its ability to collect fine-scale geospatial data due to privacy concerns, especially around minors. Blockchain algorithms, if employed fairly and transparently, can increase ownership of data and also retain privacy. Some of this technology could be implemented in low-cost, wearable sensors. This would allow for wireless devices and geospatial tracking and greatly enhance the utility of wearable sensors.

## Figures and Tables

**Figure 1 sensors-22-01295-f001:**
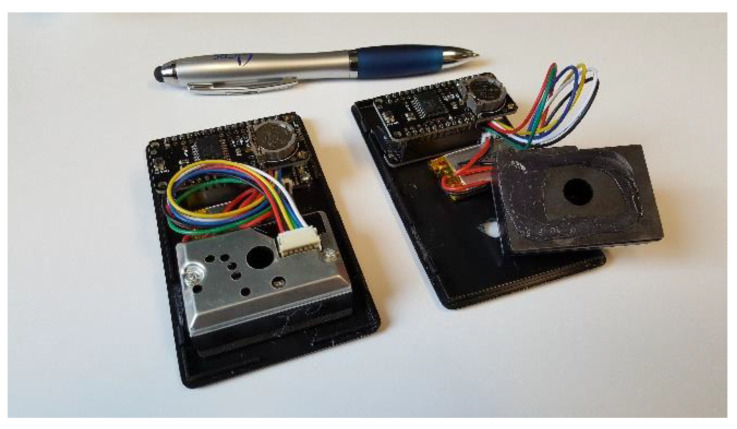
Internal design of two wearable PM sensors.

**Figure 2 sensors-22-01295-f002:**
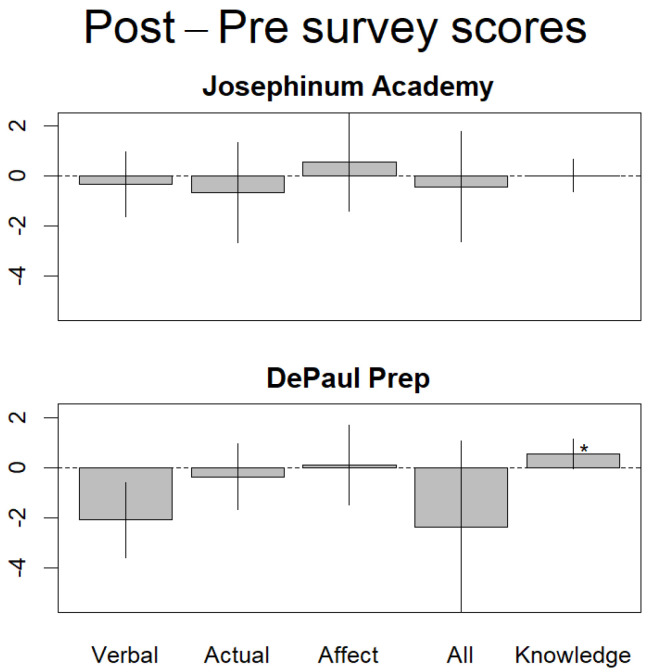
Changes of environmental attitudes sorted by assessment type. Results are based on the net change of survey scores for each attitudinal subscale. Josephinum Academy had a sample size of n = 19 responses, and DePaul Prep had a sample size of n = 9 responses. * Indicates the result would have been significant with a two-tailed *t*-test.

**Figure 3 sensors-22-01295-f003:**
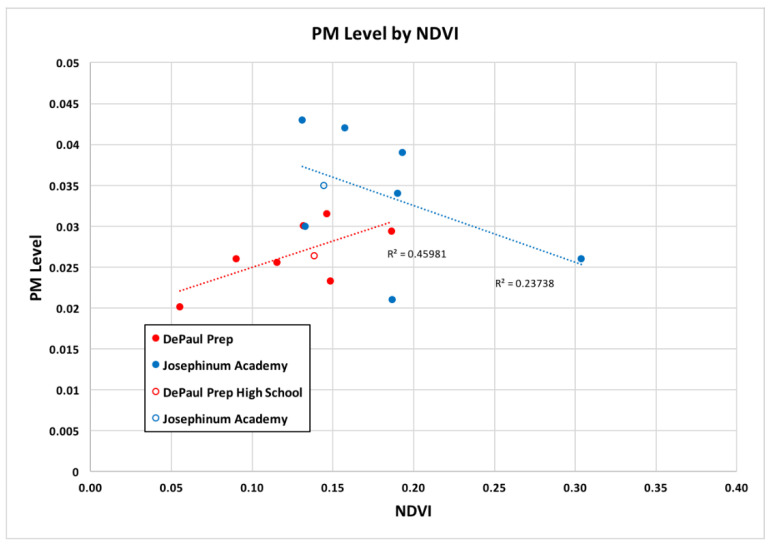
PM versus NDVI concentration for each experiment. The monitoring sites were student homes (closed circles) and each high school (open circles). The linear regression model is included for average PM concentrations and NDVI across each of the included monitoring sites (n = 7 for both Josephinum and DePaul Prep). For Josephinum, r^2^ = 0.24 and for DePaul Prep, r^2^ = 0.46.

**Figure 4 sensors-22-01295-f004:**
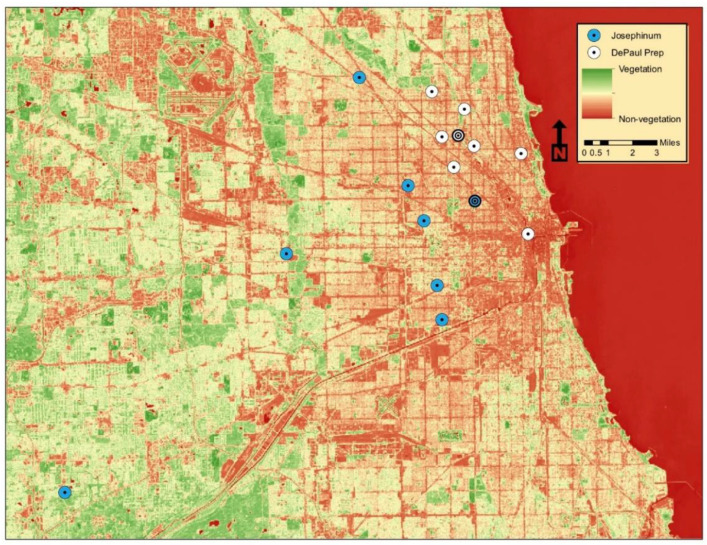
Spatial analysis of NDVI and PM concentrations. The points display the monitoring sites, colored white and blue to distinguish between DePaul Prep High School and Josephinum Academy. The NDVI layer depicts a color scale in which vegetation is green and non-vegetation is red. The Landsat 8 OLI/TIRS imagery from 12 September 2016 was acquired through the U.S. Geological Survey’s EarthExplorer. For visual purposes, we are using the Landsat image from the Josephinum experiment. The two points with concentric circles indicate the two study high schools.

**Table 1 sensors-22-01295-t001:** CHEAKS survey results. The pre- and post-experiment survey results for Josephinum Academy, with a sample size of n = 9 responses. The Likert-type attitude scores were out of a total 75 points, while the binary knowledge-based scores were out of 5.

Sensor ID	Pre Attitudes	Post Attitudes	Pre Knowledge	Post Knowledge
1	55	54	2	3
2	64	66	2	2
3	47	49	3	3
4	42	45	2	3
5	53	54	3	3
6	65	67	3	3
7	48	41	4	4
8	45	42	2	2
9	41	38	2	0
Mean	51.11	50.67	2.56	2.56
Std. Err.	2.95	3.51	0.24	0.38
*p*-value	0.65		0.50	

**Table 2 sensors-22-01295-t002:** CHEAKS survey results. The pre- and post-experiment survey results for DePaul Prep High School, with a sample size of n = 19 responses. Sensors 1 and 22 had incomplete data, and sensor 10 was missing. No pre-experiment results were significant. The Likert-type attitude scores were out of a total 75 points, while the binary knowledge-based scores were out of 5.

Sensor ID	Pre Attitudes	Post Attitudes	Pre Knowledge	Post Knowledge
2	56	41	2	3
3	64	59	4	5
4	46	56	1	3
5	48	41	5	5
6	53	56	2	3
7	39	48	3	4
8	52	48	1	3
9	54	52	3	4
11	40	46	3	3
12	64	59	4	3
13	50	34	3	3
14	59	55	1	2
15	46	43	3	3
16	59	57	4	2
17	48	37	4	2
18	60	52	2	3
19	67	64	2	4
20	44	51	3	3
21	62	67	1	3
Mean	53.21	50.84	2.68	3.21
Std. Err.	1.91	2.06	0.28	0.20
*p*-value	0.91		0.04	

## Data Availability

The data presented in this study are available on request from the corresponding author. The data are not publicly available due to conditions of the Institutional Review Board approval.

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
