# Peer review of "Monitoring Particulate Matter with Wearable Sensors and the Influence on Student Environmental Attitudes"

_sensors, 2022, doi:10.3390/s22031295_

Round 1

Reviewer 1 Report

I found some merits in this methodology and results. In my opinion, this paper has a good potential to be published in the journal. The statistical analysis performed is just enough but I find the application in this field original. However, I have also some concerns about several parts of the manuscript. In my opinion, the manuscript structure is good, but you must improve the environmental aspect of air quality in urban areas, especially on the PM. The references are many limited, so is necessary to add some specific works.  I suggest some papers be added in the references, in order to improve them and make this work more complete:

  • Air quality data for Catania: Analysis and investigation case study 2012-2013 (near line 33)

If you improve the weak points,  this manuscript will deserve to be published in this journal.

The choice of this paper derives from the lack of clarity and poorly documented situation in urban centers.  The authors have to argue the situation of air quality in urban centers.

Author Response

We would like to thank both the reviewers for the time and consideration of our original manuscript. Both noted that we needed to expand the number of references and extend the manuscript at several points. We have worked to address both of these points. We have expanded the body of the text from 336 lines to 463 lines. We have also increased the number of references from 16 to 41. But more than quantity, we have more thoroughly covered the previous literature around wearable sensors and also discussed the limitations around our experimental design. Since our research involved testing both a social science and a nature science hypothesis, we had tight constraints on our measurement technology. We wanted to work with students at a high school, and this required approval from our institutional review board. Upon reading the comments from the reviewers, we realized that we did not adequately articulate how the IRB limitations impacted our experimental design. We have added much additional text that clarifies this point. We have also greatly expanded the description of the wearable device. Finally, we have added several paragraphs in the Discussion that talk about implications of our research in light of recent air pollution controversies in Chicago and also talk about the promise of blockchain technology.

I found some merits in this methodology and results. In my opinion, this paper has a good potential to be published in the journal. The statistical analysis performed is just enough but I find the application in this field original. However, I have also some concerns about several parts of the manuscript. In my opinion, the manuscript structure is good, but you must improve the environmental aspect of air quality in urban areas, especially on the PM. The references are many limited, so is necessary to add some specific works.  I suggest some papers be added in the references, in order to improve them and make this work more complete:

Air quality data for Catania: Analysis and investigation case study 2012-2013 (near line 33)

Inserted a citation for Lanzafame et al 2014 at the end of the first paragraph.

If you improve the weak points,  this manuscript will deserve to be published in this journal.

Thanks, we appreciate your support. We understand this is a rather complex manuscript since it addresses both social and natural science hypotheses.

The choice of this paper derives from the lack of clarity and poorly documented situation in urban centers.  The authors have to argue the situation of air quality in urban centers.

We have more than doubled the number of references, and in particular discussed the health implications of poor air quality in urban centers.

Reviewer 2 Report

An interesting and practical paper, focused on monitoring the quality of air using wearable sensors. The introduction is informative, however it is far too short.

Where is the review of previously published papers, advancements in sensor networks, wireless technologies, wearable devices, etc.? All of those need to be addressed and followed by appropriate citations.

Moreover, what is the added value of this paper? It should be clearly stated and highlighted. Additionally, do mention about air pollutants, their type, effect on biological organisms and human individuals. What is their cause, how can we overcome them? Justify why is it important to study such a topic.

About the number of individuals participating in this study – was it really 9 + 19 (from the first and second school)? A group of less than 30 people is very small for a survey or questionnaire. It can be only considered as a preliminary study. What was this survey about? What were those question? Why was their number changed from 66 to 20? Additional information is needed. Information from the Appendix is not good enough.

Next, how was it conducted? Was it in a remote way (online) or stationary (a printed sheet)? How long did it last? What was the age group of participating individuals? What was their gender (male or female), background, etc.?

About the utilized sensor – was it custom-build or purchased on the market? Was it low-, medium-, or high-class? Mention something about its specification, including wireless communication interface, build-in CPU, sensors, IMUs, etc. You can provide this information in a table.

Where, how long, and at what altitude (above ground) were those measurements carried out? How many samples did you obtain from each point? Provide a digital map with measurement points marked on it. What about their nearest surroundings? Was it close to open terrain, high multistory buildings, or a street? Additionally, mark the serving area (radius) of each deployed sensor on the map).

The level and quality of the presentation of the results leaves much to be desired. With those sensor nodes you can obtain a lot more of data. Present them in an interesting way for the potential reader.

The Discussion chapter is far too short. Surely, as researchers, academics and professionals working in the field for a couple of years, you can provide more feedback, not to mention inspiration for the potential reader.

Multiple editorial and formatting issues are present.

The number of cited references is far too short. What about wireless communication, sensor networks, GIS and spatial systems, LBS services, etc.? Surely additional journal papers or conference proceedings could be easily found, analyzed and presented.

An additional Future Works chapter seems necessary. Mention about open questions and aspects that still require further investigation.

It is a pity that the Authors did not take much effort when preparing their joint paper. Taking all the above into consideration, I cannot recommend this paper in its current form. Nevertheless, Authors are encouraged to extend, modify and upgrade their manuscript and resubmit it to the journal for a second evaluation.

Author Response

We would like to thank both the reviewers for the time and consideration of our original manuscript. Both noted that we needed to expand the number of references and extend the manuscript at several points. We have worked to address both of these points. We have expanded the body of the text from 336 lines to 463 lines. We have also increased the number of references from 16 to 41. But more than quantity, we have more thoroughly covered the previous literature around wearable sensors and also discussed the limitations around our experimental design. Since our research involved testing both a social science and a nature science hypothesis, we had tight constraints on our measurement technology. We wanted to work with students at a high school, and this required approval from our institutional review board. Upon reading the comments from the reviewers, we realized that we did not adequately articulate how the IRB limitations impacted our experimental design. We have added much additional text that clarifies this point. We have also greatly expanded the description of the wearable device. Finally, we have added several paragraphs in the Discussion that talk about implications of our research in light of recent air pollution controversies in Chicago and also talk about the promise of blockchain technology.

An interesting and practical paper, focused on monitoring the quality of air using wearable sensors. The introduction is informative, however it is far too short.

Where is the review of previously published papers, advancements in sensor networks, wireless technologies, wearable devices, etc.? All of those need to be addressed and followed by appropriate citations.

We have added numerous (well over a dozen) citations and much additional text to address this point. We cite a number of different studies that used wearable devices. We also expand our discussion on why using wireless technology was not possible given our agreement to use students, including minors, as human subjects.

Moreover, what is the added value of this paper? It should be clearly stated and highlighted. Additionally, do mention about air pollutants, their type, effect on biological organisms and human individuals. What is their cause, how can we overcome them? Justify why is it important to study such a topic.

We appreciate the clear questions raised in this comment. We have added additional text to emphasize that the study was a combination of both sensor technology and social science. We have also added additional background information about air pollution impacts and corresponding references. Finally, we have contextualized our study within the framework of environmental racism that exists within the city of Chicago. We also added additional text about how in the future blockchain technology could be employed to preserve data ownership and privacy concerns associated with using community partners, and particularly minors, as human subjects. See the two new additional paragraphs at the end of the Introduction.

About the number of individuals participating in this study – was it really 9 + 19 (from the first and second school)? A group of less than 30 people is very small for a survey or questionnaire. It can be only considered as a preliminary study. What was this survey about? What were those question? Why was their number changed from 66 to 20? Additional information is needed. Information from the Appendix is not good enough.

Yes, we had a small sample size. There were 11 students and 22 students, respectively in the two high school classes. But not all students completed all elements of the experiment, so the final n is 9 and 19 for the surveys and lower for the sensors due to malfunction. We also discuss the constraints of the experimental design at the end of the Introduction and explained the small sample size.

Next, how was it conducted? Was it in a remote way (online) or stationary (a printed sheet)? How long did it last? What was the age group of participating individuals? What was their gender (male or female), background, etc.?

Due to the constraints of the Institutional Review Board, we cannot provide demographic information. We have added much additional text to describe how the experiment was conducted in the Methodology (an example included below) and at the end of the Introduction to discuss this. Also, we added additional information about the printed sheet used to record student location information.

“While there are state-of-the-art techniques to record position and wireless transmit data [16], gaining permission to do the study with high school students created challenges. Many of the participating students were minors (under age 18) and the permission of their parents was necessary. To alleviate privacy concerns, the sensors were designed with no wireless interface and no GPS or other location information. The sensors stored data on a microSD card that needed to be downloaded manually. Students recorded their movements with a low-tech paper map and pencil, a real-time clock on the device was used to synchronize concentration data with the student’s location recorded on the map with timestamps.”

About the utilized sensor – was it custom-build or purchased on the market? Was it low-, medium-, or high-class? Mention something about its specification, including wireless communication interface, build-in CPU, sensors, IMUs, etc. You can provide this information in a table.

Provided much additional information about the sensor design.

Where, how long, and at what altitude (above ground) were those measurements carried out? How many samples did you obtain from each point? Provide a digital map with measurement points marked on it. What about their nearest surroundings? Was it close to open terrain, high multistory buildings, or a street? Additionally, mark the serving area (radius) of each deployed sensor on the map).

We cannot provide any additional geographical data about student movements due to the Institutional Review Board protocol. We did modify the text that describes the experimental protocol to make some of the other above points more clear, including a note about length scale. We also included a numbered list of the steps that were taken to

The level and quality of the presentation of the results leaves much to be desired. With those sensor nodes you can obtain a lot more of data. Present them in an interesting way for the potential reader.

Again, we are limited by the IRB protocol from presenting more details of the geography. Also, the analysis was kept relatively simple so we could explain it the high school students.

The Discussion chapter is far too short. Surely, as researchers, academics and professionals working in the field for a couple of years, you can provide more feedback, not to mention inspiration for the potential reader.

Greatly expanded the Discussion. We have added several paragraphs that talk about implications of our research in light of recent air pollution controversies in Chicago and also talk about the promise of blockchain technology.

Multiple editorial and formatting issues are present.

We will be happy to resolve these as identified.

The number of cited references is far too short. What about wireless communication, sensor networks, GIS and spatial systems, LBS services, etc.? Surely additional journal papers or conference proceedings could be easily found, analyzed and presented.

Again, we have more than doubled the number of references.

An additional Future Works chapter seems necessary. Mention about open questions and aspects that still require further investigation.

Again, added the idea of using blockchain technology to address privacy and data ownership concerns.

It is a pity that the Authors did not take much effort when preparing their joint paper. Taking all the above into consideration, I cannot recommend this paper in its current form. Nevertheless, Authors are encouraged to extend, modify and upgrade their manuscript and resubmit it to the journal for a second evaluation.

We hope that are extension revisions and additions have greatly improved this manuscript. We thank the reviewer again for encouragement and direction in improving our manuscript.

Round 2

Reviewer 1 Report

In my opinion, this manuscript now has a good potential to be published in the journal

Reviewer 2 Report

Thank you for addressing to my comments and suggestions. The provided explanations and justifications are extensive and comprehensive. All is clear, I understand all aspects, including the IRB’s policy. The revised version of the manuscript has been significantly improved. It is now more informative, structured and coherent, while the research part is clear and understandable. The number of publications cited is now sufficient and appropriate.

The topic discussed is important, relevant and practical. It covers many aspects in the field of quality of life, ICT and health protection. Therefore, the paper deserves a proper and adequate representation.

In my opinion, the paper fulfils all necessary requirements in order to be accepted. I will be one of the first readers to acquaint with the final version after its publication on MDPI’s website.